# Group Fairness Meets the Black Box: Enabling Fair Algorithms on Closed LLMs via Post-Processing

## Abstract

Instruction fine-tuned large language models (LLMs) enable a simple zero-shot or few-shot prompting paradigm, also known as in-context learning, for building prediction models. This convenience, combined with continued advances in LLM capability, has the potential to drive their adoption across a broad range of domains, including high-stakes applications where group fairness—preventing disparate impacts across demographic groups—is essential. The majority of existing approaches to enforcing group fairness on LLM-based classifiers rely on traditional fair algorithms applied via model fine-tuning or head-tuning on final-layer embeddings, but they are no longer applicable to closed-weight LLMs under the in-context learning setting, which include some of the most capable commercial models today, such as GPT-4, Gemini, and Claude. In this paper, we propose a post-processing framework for deriving fair classifiers from closed-weight LLMs: we first prompt the LLM, without any fairness intervention, to produce (potentially biased) predictions on the classification task (e.g., token log probabilities or verbal elicitation) as well as predictions of group membership; we then treat these outputs as features and apply a fair algorithm to train a lightweight classifier that enforces group fairness. Experiments on five datasets, including three tabular ones, demonstrate strong accuracy-fairness trade-offs for classifiers trained on features derived from our framework; in particular, in low-resource settings, they outperform fair classifiers trained via head-tuning on LLM embeddings or from scratch on raw tabular features.[1]

## 1 Introduction

Instruction fine-tuned large language models (LLMs) such as Llama 3 (Llama Team, 2024), Gemma 3 (Gemma Team, 2025), and GPT-4 (OpenAI, 2024)—equipped with vast and diverse knowledge from their training corpora—have enabled a new paradigm, known as in-context learning, for building task-specific predictors: users can adapt these models to the task at hand by simply prompting them with the task description, often requiring little (*few-shot*) to no training data (*zero-shot*; Brown et al., 2020; Ouyang et al., 2022; Wei et al., 2022a). A common use case is classification, especially in low-resource domains like healthcare, where labeled data are expensive to acquire (Singhal et al., 2023): Hegselmann et al. (2023) demonstrate that few-shot prompting can outperform traditional machine learning (ML) models trained from scratch on tabular datasets in low-data regimes. As LLMs continue to improve in reasoning capabilities (Wei et al., 2022b), there is increasing appeal to apply this paradigm to more complex tasks in high-stakes domains such as criminal justice and finance (Barocas & Selbst, 2016; Berk et al., 2021). In applications like these, it is often essential that model predictions satisfy notions of *group fairness* to prevent disparate impacts across demographic groups. For example, Angwin et al. (2016) found that an ML-based recidivism prediction tool called COMPAS exhibited higher false positive rates for individuals from marginalized groups.

To address these concerns, the algorithmic fairness literature offers a variety of *fair algorithms* for training classifiers that satisfy group fairness criteria, and prior work has applied them to open-weight LLMs to derive fair classification models (reviewed in Section 2), typically with fine-tuning,

---

[1]Code will be immediately released after the anonymity period.

prompt-tuning, or head-tuning over last-layer embeddings. However, these approaches are no longer applicable to commercial closed-weight LLMs (e.g., GPT-4, Gemini, Claude) that do not expose model weights or internal representations, but only the output tokens and, at best, their log probabilities. So instead, recent work has explored prompt-based fairness interventions that leverage LLMs' instruction-following abilities, such as by explicitly prompting the model to "*assign labels equally across groups*" or by constructing few-shot demonstrations that reflect the fairness criterion (Liu et al., 2024; Hu et al., 2024; Li et al., 2024; Cherepanova et al., 2024). While these approaches show promise, their effectiveness can be inconsistent (e.g., due to sensitivity to prompt phrasing; Zhao et al., 2021), and the black-box nature of closed-weight LLMs means that it is unclear how to iterate on a prompt when the fairness goals are not met, beyond trial-and-error.

In this paper, we propose a framework for deriving fair classifiers from (closed-weight) LLMs with zero-shot or few-shot prompting abilities. The framework is instantiated with a fair algorithm and an LLM. It requires only access to the LLM's probabilistic predictions (e.g., log probabilities over label tokens), without any model tuning or embedding access. It treats the LLM as a feature extractor, and elicits useful features from its probabilistic predictions via prompting; then, the chosen algorithm is applied to train a (lightweight) fair classifier on the extracted features (which, ideally, contain sufficient statistics for fair classification). Our framework is broadly applicable to closed-weight LLMs, and supports a wide range of group fairness criteria under both *attribute-aware* and *attribute-blind* settings, where the sensitive attribute (group membership) is not observed at test time.

We empirically evaluate our framework on five datasets, instantiated with four LLMs and three fair algorithms. Three of the LLMs are open-weight models, allowing us to compare our framework against the same fair algorithms applied to LLM embeddings. Additionally, three of the datasets are tabular, allowing comparisons with training on raw tabular features. Across datasets, fairness criteria, and both open-weight and closed-weight LLMs, our framework consistently produces fair classifiers with strong accuracy tradeoffs. Further experiments under varying training set sizes highlight its data efficiency: in low-data regimes, our framework outperforms fair classifiers trained on LLM embeddings or from scratch on tabular features, since the low-dimensional features it extracts reduce the sample complexity of learning—these findings complement those of Hegselmann et al. (2023), who show that LLMs achieve competitive few-shot performance on tabular classification tasks, by demonstrating that they are similarly competitive in fair classification.

## 2 RELATED WORK

We begin with a brief overview of the algorithmic fairness literature, followed by a review of prior work on group fairness in LLMs. While this work focuses on group fairness in classification problems, other fairness notions have been proposed for different contexts. For example, *subgroup fairness* considers finer-grained group definitions (Hébert-Johnson et al., 2018; Kearns et al., 2018), *individual fairness* requires that similar individuals receive similar predictions (Dwork et al., 2012); Chzhen et al. (2020) and Le Gouic et al. (2020); Zhao (2023) study fairness in regression problems.

Fairness concerns extend beyond predictive models: for (generative) language models, a line of work beginning with Bolukbasi et al. (2016) and Caliskan et al. (2017) investigates the social biases inherited from pre-training corpora. Since then, numerous benchmarks and evaluation protocols have been developed to assess harms in language generation (May et al., 2019; Nangia et al., 2020; Nadeem et al., 2021; Shaikh et al., 2023; Bai et al., 2024). For example, the BBQ benchmark (Parrish et al., 2022) has been used for bias evaluation in recent GPT releases (OpenAI, 2024). While such biases may influence the fairness of LLM predictions on downstream classification tasks, it remains unclear whether mitigating them directly improves group fairness, which is a statistical notion that depends on the task's data distribution. We therefore view this body of research as orthogonal to efforts, such as ours, that specifically aim to enforce group fairness on downstream tasks.

**Fair Algorithms.** Algorithms for training classifiers that satisfy group fairness in traditional ML settings can be categorized by the stage at which mitigation occurs. Pre-processing methods clean the training data to remove biased associations, using techniques such as reweighting and subsampling (Kamiran & Calders, 2012; Calmon et al., 2017). In-processing methods incorporate fairness constraints directly into the model's training objective (Zafar et al., 2017); notable examples include fair representation learning (Zemel et al., 2013; Prost et al., 2019; Zhao et al., 2020) and the Re-

ductions approach (Agarwal et al., 2018). Post-processing methods adjust the model's outputs after training to enforce fairness (Menon & Williamson, 2018; Alghamdi et al., 2022; Ţifrea et al., 2024; Xian & Zhao, 2024). Many of these algorithms provide theoretical guarantees on fairness, provided that sufficient training examples are available and certain assumptions hold.

**Applying Fair Algorithms on LLMs.** To derive fair classifiers from LLMs, prior work has combined fair algorithms above with some form of (parameter-efficient) tuning. However, due to the cost of fine-tuning and the need to access model weights or internal representations (e.g., last-layer embeddings for head-tuning), these approaches have primarily been applied to small, open-weight models such as BERT (Devlin et al., 2019), GPT-2 (Radford et al., 2019), and Llama. From the pre-processing category, Han et al. (2022) reweights the training data and performs head-tuning, while Garg et al. (2019) and Fatemi et al. (2023) use counterfactual data augmentation to balance group-label distributions, followed by fine-tuning or prompt-tuning. In the in-processing category, Han et al. (2021) learn fair representations over embeddings while freezing the transformer layers, and Cherepanova et al. (2024) incorporates a fairness regularization term into prompt-tuning.

Post-processing approaches have also been explored. Atwood et al. (2025) apply the post-processing algorithm from Ţifrea et al. (2024) to learn a fair classification head (i.e., head-tuning). Baldini et al. (2022) considers equalized odds in the attribute-aware setting and applies algorithms from Hardt et al. (2016) and Alghamdi et al. (2022) to the predictions of LLMs (that were previously fine-tuned on downstream tasks), without additional tuning. This setup is closely related to ours in that it requires only access to LLM predictions and performs post-hoc fairness mitigation, but it is limited to the attribute-aware setting. Our framework generalizes this approach by supporting a broader range of fairness criteria and the attribute-blind setting. We also evaluate in-processing algorithms in addition to post-processing within our framework.

Beyond fair algorithms and prompt-based mitigation (discussed in the introduction), several (unsupervised) methods have been proposed to improve group fairness in downstream tasks, including methods based on null-space projection that remove group-sensitive information from LLM embeddings (Bordia & Bowman, 2019; Ravfogel et al., 2020), as well as more recent embedding steering techniques (Singh et al., 2024).

## 3 PRELIMINARIES

We consider a classification task defined by a joint distribution over inputs $X$ (e.g., text or tabular data), class labels $Y \in \{1, \ldots, K\}$, and sensitive attributes $A \in \{1, \ldots, G\}$ representing demographic groups (e.g., sex, race). Given labeled examples $\{(x_i, y_i, a_i)\}_{i=1}^{N}$, the goal is to learn a classifier that satisfies group fairness criteria.

**LLM Predictions.** Rather than training fair classifiers from scratch, we aim to use pre-trained LLMs as base prediction models, leveraging their ability to answer classification queries via zero-shot or few-shot prompting. To obtain predictions from an LLM, we design a prompt template with task instructions in a multiple-choice question-answering (MCQA) format, where each answer choice corresponds to a class label. Each example is inserted into the prompt, and the LLM is queried to obtain log probabilities (logits) over the output tokens associated with each class. These serve as the model's probabilistic predictions for the class labels.[2]

Before using these predictions (e.g., taking the argmax), it is advisable to re-fit or calibrate them on labeled examples, such as via logistic regression. This is because the raw predictions may perform poorly due to distribution shifts between the LLM's pre-training distribution (or internal beliefs) and the distribution of the task. They may also exhibit sensitivity to prompt phrasing and formatting (Zhao et al., 2021), further motivating the need for calibration.

**Group Fairness.** Group fairness is a statistical notion that examines disparities in the output distribution of a classifier $\widehat{Y}$ conditioned on the sensitive attribute $A$. We focus on four standard group fairness criteria, and define the corresponding measure of fairness violation below:

---

[2]If token-level logits are not accessible, alternative approaches include sampling multiple completions under non-zero temperature (Cecere et al., 2025) or using *verbal elicitation* (Xiong et al., 2024).

- **Statistical Parity** (SP; Calders et al., 2009). Requires the classifier's output distribution to be approximately equal across all groups $a \in [G]$:

$$V_{\text{SP}} = \max_{a,a' \in [G],\, k \in [K]} \left| \mathbb{P}(\widehat{Y} = k \mid A = a) - \mathbb{P}(\widehat{Y} = k \mid A = a') \right|.$$

- **True Positive Rate Parity** (TPR; Hardt et al., 2016). Defined for binary classification ($K = 2$), it requires the true positive rate to be equal across groups:

$$V_{\text{TPR}} = \max_{a,a' \in [G]} \left| \mathbb{P}(\widehat{Y} = 2 \mid A = a, Y = 2) - \mathbb{P}(\widehat{Y} = 2 \mid A = a', Y = 2) \right|.$$

  **False Positive Rate Parity** (FPR) is defined analogously by conditioning on $Y = 1$.

- **Equalized Odds** (EO; Hardt et al., 2016). Requires all types of classification error to be balanced across groups:

$$V_{\text{EO}} = \max_{a,a' \in [G],\, j,k \in [K]} \left| \mathbb{P}(\widehat{Y} = k \mid A = a, Y = j) - \mathbb{P}(\widehat{Y} = k \mid A = a', Y = j) \right|.$$

Generally, a group fairness criterion seeks to equalize the distribution of classifier outputs $\widehat{Y}$ conditioned on $(A = a, B = b)$ for some auxiliary variable $B$, across different groups $a$. For example, TPR, FPR, and EO fairness set $B = Y$, reflecting the true qualification of the individual, whereas SP does not use $B$.

Last but not least, we distinguish between two settings: in the *attribute-aware* setting, the sensitive attribute $A$ is available at test time and may be used for prediction or mitigation; in *attribute-blind*, $A$ is not observed at test time. This work focuses on the more general attribute-blind setting.

# 4 A FRAMEWORK FOR DERIVING FAIR CLASSIFIERS FROM LLMS VIA POST-PROCESSING

We describe a framework for deriving fair classifiers from LLMs using zero-shot or few-shot prompting. The framework is instantiated with an LLM (possibly closed-weight) and a fair algorithm. It treats the LLM as a frozen feature extractor: features are derived from the LLM's probabilistic predictions on strategically designed prompts, which are then passed to the fair algorithm to train a lightweight fair classifier using labeled examples. While such a setup may appear unnecessary for open-weight LLMs, e.g., one can directly use last-layer embeddings for head-tuning, this is not feasible with closed-weight models. Thus, the novelty of our framework lies in the design of the post-processing pipeline that uses the elicited informative features purely from the LLM's output predictions; so the question becomes: *What information about the input $X$ should the LLM summarize—in the form of predictions elicited from prompts—to support fair classification?*

Our proposal is motivated by recent work on post-processing algorithms for group fairness (Chen et al., 2024; Zeng et al., 2024), which shows that the Bayes-optimal fair classifier for a given task can be expressed as a (potentially randomized) function of the conditional distribution of the task labels $\mathbb{P}(Y \mid X)$ and the joint $\mathbb{P}(A, B \mid X)$, where $(A, B)$ are the variables conditioned on in the fairness criterion; moreover, under mild continuity conditions on the data distribution, this function reduces to a simple deterministic mapping (e.g., linear; Xian & Zhao, 2024). These results imply that if we can prompt the LLM to produce (accurate estimates of) $\mathbb{P}(Y \mid X = x)$ and $\mathbb{P}(A, B \mid X = x)$ for each input $x$, then these predictions constitute sufficient statistics for (optimal) fair classification, and can be used as features to train fair classifiers.

**Proposed Framework.** The framework consists of three steps: (1) design prompt template(s) for eliciting LLM predictions for $Y$ and $(A, B)$ given the input, depending on the fairness criterion; (2) prompt the LLM with each input inserted into the template(s); and (3) construct features from the predictions and apply the fair algorithm to learn a fair classifier. Figure 1 provides a flowchart using the ACSIncome task and EO fairness as an example (Section 5.1; see Table 3 for an example input). Each step is elaborated below:

1. *Design prompt(s).* For EO fairness (where $B = Y$), we need to obtain predictions for $\mathbb{P}(A, Y \mid X)$. Note that there is no need to separately predict $Y$, since $\mathbb{P}(Y \mid X) = \sum_a \mathbb{P}(A = a, Y \mid X)$.[3]

---

[3] For SP, TPR, FPR, and EO fairness in the attribute-aware setting, only $\mathbb{P}(Y \mid X)$ is needed, since $A$ is always observed. This reduces to the setting of Baldini et al. (2022).

**Task** (ACSIncome). Predict if the annual income is >\$50k ($Y$). The sensitive attribute $A$ is race.

**Equalized Odds.** $\Pr(\hat{Y} = k \mid A = a, Y = j) = \Pr(\hat{Y} = k \mid A = a', Y = j)$ for all groups $a, a'$ and labels $j, k$.

1. Design prompt(s) to predict $(A, Y)$    2. Prompt for LLM predictions    3. Featurize & apply fair algorithm

(here, decomposed into predicting $Y$ and $(A \mid Y = j)$ for all $j$)

Figure 1: Flowchart of our proposed framework for deriving fair classifiers from LLMs. Given a task and a fairness criterion, we first design prompts to elicit LLM predictions for the task label $Y$ and variables associated with the criterion. These predictions are collected for each training example, transformed into feature vectors, and fed into a fair algorithm to train a lightweight fair classifier.

Since both $A$ and $Y$ are categorical, one option is to construct a single MCQA prompt with $G \times K$ choices. For instance, on ACSIncome: "*A. The income is >\$50k and the race is White*", "*B. The income is >\$50k and the race is Black*", and so on. Alternatively, we may factor the joint distribution as $\mathbb{P}(A, Y \mid X) = \mathbb{P}(Y \mid X)\,\mathbb{P}(A \mid Y, X)$ and design $(1 + K)$ prompts accordingly: one to predict $Y$, and the rest to predict $A$ conditioned on hypothetical values of $Y$, e.g., "*Given the income is >\$50k, what is the race?*" (see Listing 1).[4] This decomposition simplifies each prediction task and could lead to better LLM predictions, though at the cost of multiple queries per example.

On tasks where $A$ can be easily inferred from $X$ (e.g., BiasBios and CivilComments datasets; see X for examples), the conditional independence assumption $\mathbb{P}(A, Y \mid X) = \mathbb{P}(A \mid X)\,\mathbb{P}(Y \mid X)$ may approximately hold. In such cases, if prompt decomposition is used, we can leverage this structure and reduce the number of prompts to two: one for predicting $Y$ and the other for $A$ (see Listings 4 and 5).

2. *Prompt for LLM predictions.* For each input $x$, we insert it into the designed template(s) and query the LLM (over multiple messages). We extract the predictions by taking only the logits of the output tokens corresponding to the MCQA option choices (each mapped to a label; see Footnote 2 for alternatives when logits not accessible). If only the top-$k$ logits are accessible (for instance, $k = 20$ in OpenAI's GPT-4o API at the time of writing, due to intellectual property protections; Carlini et al., 2024) and not all option choices in the prompt template appear (particularly when prompting for label predictions on the 28-class BiasBios dataset), we assign a large negative value to the missing logits (we use $-50$ for GPT-4o).

3. *Featurize and apply fair algorithm.* The LLM predictions are transformed into feature vectors, and used to train a fair classifier with the chosen fair algorithm on labeled examples, $\{x_i, a_i, b_i, y_i\}_{i=1}^{N}$. The LLM remains frozen throughout.

   Feature engineering can be as simple as concatenating the raw logits, or involve more elaborate procedures: in our implementation, we first aggregate and reshape the logits into a $G \times K$ vector that semantically encodes the LLM's prediction of the full joint $\log \mathbb{P}(A, Y \mid X)$ (see Appendix D for the exact formula), then fit these to the ground-truth $(A, Y)$ labels using logistic regression for calibration (these training data are shared with the fair algorithm), and finally, use the calibrated probabilities as features (lying in the $\Delta^{G \times K}$ space) for the fair algorithm.

At test time, we use the prompt(s) from step (1) to obtain predictions from the LLM, featurize them as in step (3), then call the trained fair classifier to make the final prediction.

---

[4]The joint can also be factored as $\mathbb{P}(A, Y \mid X) = \mathbb{P}(A \mid X)\,\mathbb{P}(Y \mid A, X)$, leading to $(1 + G)$ decomposed prompts. We used the alternative factorization in our experiments because it required fewer prompts for the datasets considered.

## 5 EXPERIMENT SETUP

We evaluate our proposed framework as described in Section 4 on three tabular and two textual datasets, using three fair algorithms, three open-weight LLMs (Llama 3.1 8B, 70B, Gemma 3 27B) and two closed-weight LLM (GPT-4o, o3).[5] For simplicity, we use zero-shot prompting to elicit LLM predictions, though few-shot or chain-of-thought prompting may be used to improve performance, or be combined with prompt-based mitigation methods reviewed in Section 1. We consider four fairness criteria: SP, TPR, FPR, and EO. For the first three, although our framework only requires partial information from the joint distribution of $(A, Y) \mid X$, we nonetheless prompt for the full joint as required by EO, so that the same setup can be used across all experiments.

**Fair Algorithms.** We instantiate our framework with two in-processing algorithms and one post-processing algorithm. For in-processing, we use **Reductions** (binary classification only; Agarwal et al., 2018) applied with logistic regression, and **MinDiff** (Prost et al., 2019), a distribution-matching approach implemented with one-hidden-layer MLP. For post-processing, we use **Linear-Post** (Xian & Zhao, 2024) and apply it directly to the re-fitted conditional probabilities of $(A, Y)$ obtained in step (3) of feature engineering. To explore the accuracy-fairness tradeoff, we sweep the fairness tolerance parameters for Reductions and LinearPost, and the regularization strength for MinDiff. Lastly, the **no-mitigation** baseline is from re-fitting the LLM's prediction of $Y$ using logistic regression without fairness mitigation. Further implementation details and hyperparameters are provided in Appendix E.

**Comparisons.** The features extracted in our framework are very low-dimensional—of $KG$ dimensions, e.g., 4 on Adult, and 10 on ACSIncome—compared to the size of LLM embeddings (4096 in Llama 3.1 8B) or the original input space (97 tabular features on Adult after pre-processing, 810 on ACSIncome). This dimensionality reduction could potentially discard significant information about the input and degrade performance.

To assess this gap, we compare our framework (**preds**) against the alternatives: applying the fair algorithms to the mean-pooled embeddings from open-weight LLMs (**embeds**),[6] and (on tabular datasets) directly to the original tabular features, training classifiers from scratch (**tabular**). In these settings, Reductions and MinDiff are applied same as above, and for LinearPost, we follow a "pre-train then post-process" procedure: first, train a logistic regression model to predict $(A, Y)$ from either the embeddings or tabular features, then, apply LinearPost to its predictions (both steps use the same training data). The no-mitigation baseline fits the same features to $Y$ using logistic regression.

### 5.1 DATASETS

We evaluate our framework on five datasets, including three tabular datasets commonly used in the fairness literature. Tabular data are converted into textual format following *list serialization* of Hegselmann et al. (2023), and we further replace the original column and category codings (usually undescriptive) with their natural language descriptions. Examples from each dataset are shown in Table 3, and additional details on pre-processing and split sizes are provided in Appendix B.

- **Adult & ACSIncome** (Kohavi, 1996; Ding et al., 2021). The task is to predict whether the annual income of an individual is over $50k (binary classification), and the sensitive attribute is sex (binary) for Adult and race (five categories) for ACSIncome. We consider SP, TPR, and EO fairness, and drop both sex and race columns from the input (attribute-blind setting). Prompts for predicting $(A, Y)$ given $X$, decomposed into separate predictions for $Y$ and $A \mid Y$, are in Listings 1 and 2, along with those for obtaining LLM embeddings.

- **COMPAS** (Angwin et al., 2016). Predict whether an individual will reoffend (recidivism; binary classification), with race (African-American or Caucasian) as the sensitive attribute. We consider SP, FPR, and EO fairness, and remove both sex and race from the input. Prompts for LLM predictions and embeddings are in Listing 3.

---

[5]The names of the model checkpoints are `Llama-3.1-8B`, `Llama-3.1-70B`, `gemma-3-27b-it` (Hugging Face), and `gpt-4o-2024-08-06`, `o3-2025-04-16` (OpenAI API).

[6]In our preliminary experiments, mean-pooling outperformed both last-token and max-pooling.

- **BiasBios** (De-Arteaga et al., 2019). Given a biography, the task is to classify the person's occupation (28-way classification). The sensitive attribute is sex (binary), which is not explicitly given but often inferable from the biography (e.g., based on the pronouns; so in the prompt design, we assume conditional independence $A \perp Y \mid X$). We consider SP and EO fairness. Prompts for LLM predictions and embeddings are in Listing 4.

- **CivilComments** (Borkan et al., 2019). The task is to detect whether a public comment is toxic (binary classification), and the sensitive attribute is the religion(s) mentioned in the comment (Christianity, Judaism, Islam, Hinduism, Buddhism). We consider FPR parity for fairness. Unlike previous tasks, since comments can mention multiple religions simultaneously (or none), the groups here are overlapping (i.e., the sensitive attribute is multi-label, $A \in \{0, 1\}^5$). Hence, we use five separate prompts to elicit LLM predictions for each religion (Listing 5; assuming conditional independence).

## 5.2 EVALUATION PROTOCOL

We define an *experiment configuration* as a combination of an LLM, a fair algorithm, and one of three types of feature used for the fair algorithm: **preds** (LLM predictions, i.e., proposed framework), **embeds** (LLM embeddings), or **tabular** (the original tabular features). This leads to at most 24 configurations per dataset (excluding no-mitigation baselines). Each configuration is run five times with different random seeds. For each task and configuration, we obtain a collection of classifiers that span various accuracy-fairness tradeoffs, and retain only those that are Pareto-optimal based on the validation set. Fairness violation is computed as defined in Section 3; for CivilComments, we extend the FPR violation to overlapping groups and apply a weighted variant (Eq. (3) in the appendix) to improve statistical power, due to the small sample size of intersectional groups.

**Area Under the Tradeoff Curve** (AUTC). To enable quantitative comparisons across the many configurations evaluated, we introduce a metric called *area under the tradeoff curve*, which summarizes each curve into a single value. AUTC is defined relative to the problem's base rate (i.e., accuracy of majority-class constant classifier, which is trivially fair under the group fairness criteria in Section 3). It computes the area under the tradeoff curve and above a horizontal line at the base rate, and normalizes this value to lie between 0 (less than 0 means performance is worse than constant classifier) and 1 (perfect fairness and accuracy, although may not be attainable; Zhao & Gordon, 2022). However, high AUTC scores can be achieved through high accuracy alone even if tradeoffs in the high-fairness (low-violation) regime are poor; this is because tradeoff curve is extended to the right, giving an advantage to high-accuracy classifiers by allowing them to accumulate more area in high-violation region. To address this, we refine AUTC in Appendix C.1 and Fig. 6 to place greater emphasis on high-fairness regime.

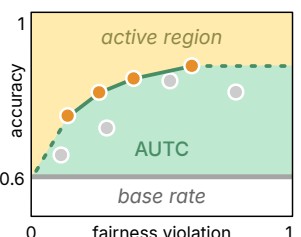

Figure 2: AUTC, in its basic form, is the area under Pareto-optimal tradeoffs within the active region (i.e., above the base rate).

## 6 RESULTS AND DISCUSSIONS

In our main experiments (Section 6.1), we evaluate our framework across datasets, fairness criteria, LLMs, and fair algorithms, and compare the utility of the features extracted from the LLM in our framework to LLM embeddings (for open-weight models) and the original tabular features. We also include two sets of comparison and ablation studies. In Section 6.2, we evaluate performance under varying training set sizes. In Appendix A, we compare our framework, which prompts the LLM for predictions of both the label $Y$ and the fairness-related variables $(A, B)$, to a variant that only prompts for $Y$, i.e., applying fair algorithms directly to the LLM's label predictions. This ablation highlights the importance of explicitly including group information in the features for effective fairness mitigation.

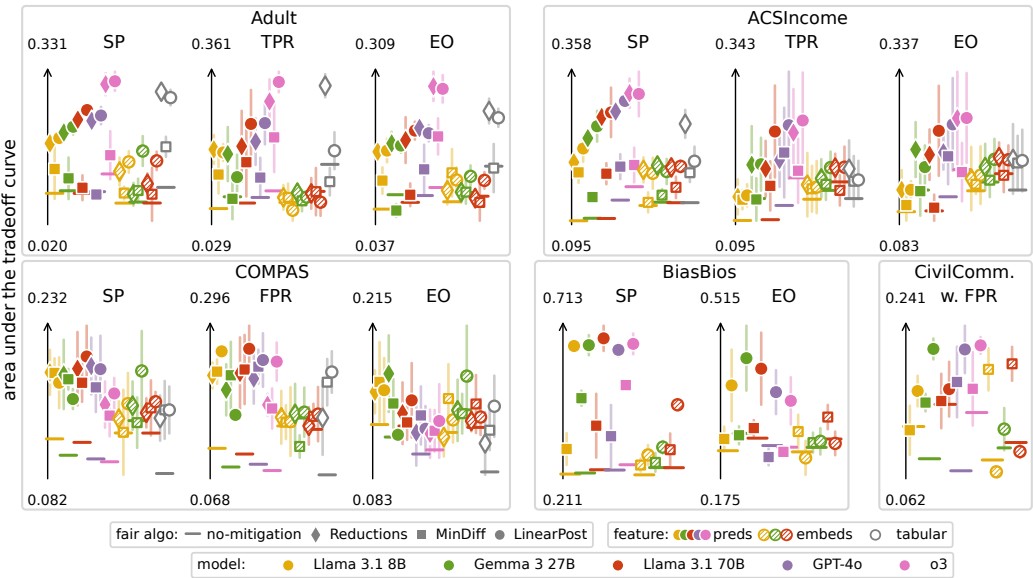

Figure 3: Area under the tradeoff curve (Appendix C.1) achieved by each configuration (Section 5.2). Our proposed framework corresponds to the **preds** feature; **embeds** refers to training fair classifiers on open-weight LLM embeddings, and **tabular** refers to training on the datasets' original tabular features. Full tradeoff curves are shown in Figs. 7 to 11 in the appendix.

## 6.1 MAIN RESULTS

In Fig. 3, we show the AUTC scores achieved by each configuration; full tradeoff curves are deferred to Figs. 7 to 11 in the appendix. Note that in this main set of experiments, we deliberately use a smaller training set size than is typical for these datasets (Table 1), because of two considerations: to replicate the data-scarce scenarios that are the primary use case for LLM prompting, and to accommodate limited compute resources.

First, as a sanity check, we find that fair algorithms generally achieve better accuracy-fairness tradeoffs than the no-mitigation baseline, which linearly interpolates between the unconstrained classifier trained on the respective features and the majority-class constant classifier. Next, we observe that our proposed framework is effective across all settings and both open-weight and closed-weight models, demonstrating its generality and soundness: despite the low dimensionality of the features it extracts, it retains the essential information needed for fair classification. Finally, and as expected, performance tends to improve with the capability of the underlying LLM.

For open-weight models, fair classifiers trained on features elicited via our framework achieve higher AUTC scores than those trained on LLM embeddings, and in some cases, even outperform those trained on the original tabular features. This can be largely attributed to the lower sample complexity of our framework, owing to the low dimensionality of the extracted features. For example, on the Adult dataset, our **preds** feature has only 4 dimensions, whereas Llama 3.1 8B embeddings have 4096—exceeding the training set size of 2000 and potentially leading to overfitting. Since LLM output logits are linearly probed from the embeddings, our framework can be conceptually viewed as a form of dimensionality reduction, selecting only the most informative components for fair classification. We further explore the impact of training set size in Section 6.2.

## 6.2 COMPARISON ACROSS TRAINING SET SIZES

In Fig. 4, we compare the AUTC scores of fair classifiers trained on features extracted via our framework, to those trained on LLM embeddings and original tabular features, across varying training set sizes on the Adult dataset using Llama 3.1 open-weight models. We focus our discussion on results for SP and EO fairness only, as TPR results exhibit high variance.

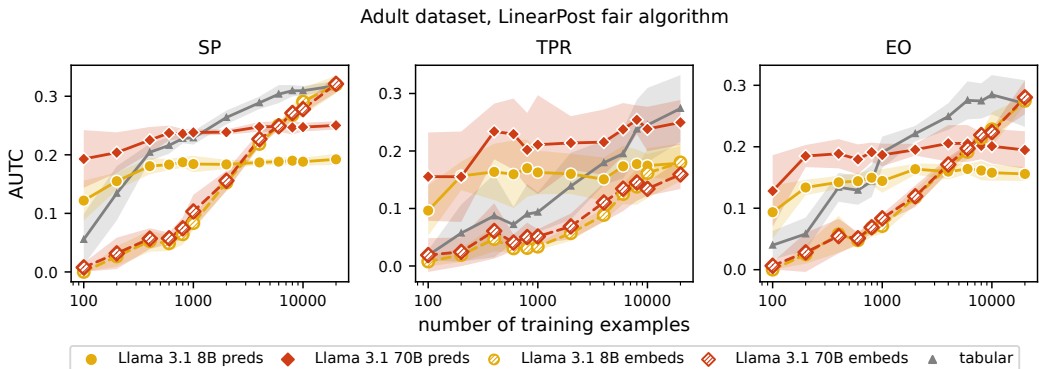

Figure 4: AUTC scores on the Adult dataset using LinearPost and Llama 3.1 open-weight models, comparing our proposed framework (**preds**) to fair classifiers trained on LLM embeddings (**embeds**) and original tabular features (**tabular**), across training set sizes.

The results reveal three distinct regimes, each favoring a different feature type, likely driven by the sample complexity induced by feature dimensionality. In the very low-data regime (fewer than 1000 examples), our framework outperforms the others due to its low dimensionality (4 on Adult), which reduces sample complexity. However, this simplicity also imposes an information bottleneck, leading to a performance plateau as training size increases. In contrast, both tabular features (97 dimensions) and embeddings (4096 for 8B and 8192 for 70B) initially perform poorly in this regime, likely due to overfitting, as their dimensionality far exceeds the number of training examples.

As the training set size increases, performance improves for both tabular features and embeddings. Tabular features outperform others in the low to mid-data regime, but are eventually overtaken by embeddings in the high-data regime (although intriguingly, the embeddings from the 8B and 70B models yield nearly identical performance).

The strong performance of our procedure in the very low-data regime highlights the value of zero-shot and few-shot prompting for classification when training data is scarce. Whereas Hegselmann et al. (2023) showed that LLM-derived classifiers can outperform traditional ML models trained from scratch on tabular features in few-shot settings, our results extend this finding by demonstrating that LLM predictions are also advantageous for *fair* classification in such low-data scenarios.

# 7 CONCLUSION

In this paper, we proposed and evaluated a framework for deriving fair classifiers from (closed-weight) LLMs by featurizing the LLM's predictions for the label and fairness-related variables (e.g., sensitive attributes), and then applying traditional fair algorithms to train lightweight classifiers on the extracted features.

While our experiments focused on zero-shot prompting, the framework naturally supports other prompting techniques such as few-shot and chain-of-thought prompting, which may further improve performance. Prompt-based fairness mitigation strategies could also be incorporated, with the fair algorithm complementing these methods by addressing residual disparities that the prompting alone fails to mitigate.

A limitation of the proposed framework is that it relies on the condition that the elicited predictions— i.e., of the label and fairness-related variables—contain sufficient information for fair classification. This condition holds only if the LLM produces Bayes-optimal predictions, which is unlikely in practice. In such cases, there could be additional information to elicit from the LLM that may improve the performance of the subsequent fair algorithm, particularly given the performance plateau observed in Fig. 4, in contrast to the continued gains from training on LLM embeddings.

NOTE ON LLM USAGE IN PAPER WRITING

The writing of this paper was assisted by OpenAI's GPT model, limited to grammar correction and sentence refinement.

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

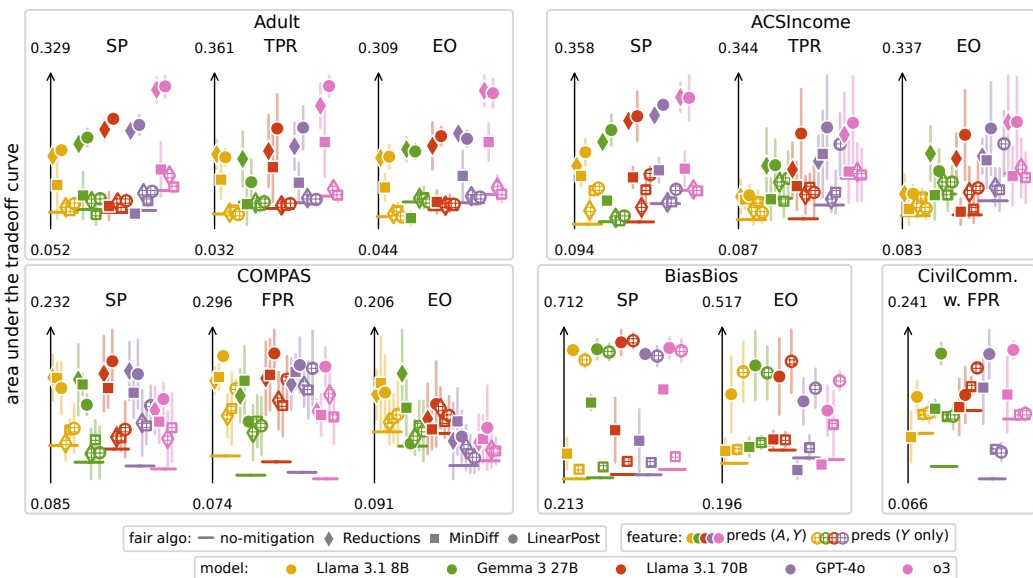

Figure 5: AUTC scores comparing our proposed framework, which prompts the LLM for both group and label predictions $(A, Y)$, to a variant that only prompts for the label, i.e., training the fair classifier on the LLM's label predictions alone.

## A  ABLATION EXPERIMENTS ON GROUP PREDICTIONS

Last but not least, in our framework, we prompt the LLM not only for predictions of the label $Y$ but also for fairness-related variables that encode group information (e.g., the sensitive attribute $A$). These predictions serve as sufficient statistics for fair classification and are used to construct features on which the fair algorithm is subsequently applied. This setup naturally raises a curious question: *are predictions of the fairness-related variables actually necessary?*

To explore this, we conduct an ablation study in which we repeat the experiments in Section 6.1, comparing our framework (which prompts for both the sensitive attribute $A$ and label $Y$), to a variant that prompts only for $Y$, meaning that the fair algorithm is applied to the LLM's predictions of the label alone.

Figure 5 presents the results. We observe that, in nearly all cases, excluding group information (i.e., omitting predictions of the sensitive attribute $A$) degrades the performance of the final derived classifier. However, there are notable exceptions, specifically on the COMPAS, BiasBios, and Civil-Comments datasets, where the exclusion does not harm performance (and can even lead to slight improvements).

Further analysis reveals two possible explanations for these exceptions. First, if the LLM performs poorly at predicting $A$, the group predictions provide little to no value. This is the case for COMPAS, where a logistic regression model trained to predict $A$ from the raw LLM log probabilities of $(A, Y)$ (constructed as described in Appendix D) achieves a balanced accuracy of only $0.6560 \pm 0.0055$. Second, if the LLM's predictions of $Y$ already encode substantial information about $A$, then prompting for $A$ offers limited additional benefit. This is observed in BiasBios, where the balanced accuracy of predicting $A$ from the log probabilities of $Y$ alone (via a fitted logistic regression model) is already $0.8768 \pm 0.0008$, compared to $0.9961 \pm 0.0001$ when using the full $(A, Y)$ log probabilities. The high dimensionality of $Y$ in BiasBios (28 classes) likely contributes to this implicit encoding.

## B  DATASETS AND PRE-PROCESSING

For the main experiments in Section 6.1, all datasets are randomly split into train, validation, and test sets as shown in Table 1. If a dataset includes a pre-defined split (e.g., Adult and BiasBios), we

Table 1: Dataset split sizes used in the experiments.

| Dataset | Train | Validation | Test |
|---|---|---|---|
| Adult (Section 6.1 and Appendix A) | 2000 | 2000 | 20000 |
| Adult (Section 6.2) | 100–20000 | 5000 | 20000 |
| ACSIncome | 10000 | 10000 | 20000 |
| COMPAS | 2000 | 1000 | 2000 |
| BiasBios | 20000 | 20000 | 50000 |
| CivilComments | 20000 | 20000 | 50000 |

merge all splits and re-split randomly. Due to resource constraints, we do not use the full dataset in every case: for example, we sample 40000 out of 1664500 examples from ACSIncome, 90000 out of 393423 from BiasBios (the version scrapped and hosted by Ravfogel et al. (2020)), and 90000 out of 447998 from CivilComments (from TensorFlow Datasets[7]). Examples from each dataset are shown in Table 3.

For the experiments in Section 6.2 that examine performance on the Adult dataset under varying training sizes, we fix the test set size to 20000, validation to 5000, and vary the training set size across 100, 200, 400, 600, 800, 1000, 2000, 4000, 6000, 8000, 10000, and 20000 examples.

For CivilComments, we label an example as toxic if any annotator marked it so (that is, if `toxicity > 0`). Similarly, a comment is considered to mention a religion if any annotator labels so (i.e., `christian`, `jewish`, `muslim`, `hindu`, or `buddhist > 0`, respectively).

**Tabular Data Serialization.** We serialize the tabular features of the tabular datasets (ADULT, ACSINCOME2-RACE, COMPAS) into a text format suitable for LLMs, following the *list serialization* approach of Hegselmann et al. (2023). Each serialized example is a multiline string where each line has the form "`{column_name}: {value}`" (columns with missing values are omitted).

For categorical features, prior work applying LLMs to tabular datasets (including studies on LLM group fairness; Liu et al., 2024; Cherepanova et al., 2024) typically populate the placeholders `{column_name}` and `{value}` with raw dataset encodings. For example, in ADULT, the line "`workclass: State-gov`" means that the individual's "Class of Worker" is "State government employee". However, these codings are often terse or opaque, especially in ACSINCOME, where, for instance, "Class of Worker" is coded as "`COW`", which may be difficult for an LLM to interpret without a dictionary.

To address this, we replace raw categorical encodings with their natural language descriptions (Li et al. (2024) uses a similar approach, who explicitly provided natural language descriptions for categorical encodings in prompts). In our preliminary experiments on ADULT, substituting raw codes with natural language descriptions improved downstream performance after re-fitting. An example with and without this substitution is shown in Table 3, and the full code-to-description mappings are available in our released code.

When training classifiers directly on TABULAR features from scratch, we apply standard pre-processing: categorical features are one-hot encoded, and all features are standardized.

**Prompt Templates.** The prompt templates used in our experiments are shown in Listings 1 to 5, each example in the dataset is inserted by filling in the placeholder `{example}`. Since the re-fitting step reduces the LLM's sensitivity to prompt phrasing, we did not devote significant effort to prompt engineering or optimization—beyond verifying that the model follows the MCQA instruction and outputs a valid option. The template used for predicting class labels on the CivilComments dataset is adapted from Atwood et al. (2025).

---

[7]Documentation is at: https://www.tensorflow.org/datasets/catalog/civil_comments#civil_commentscivilcommentsidentities.

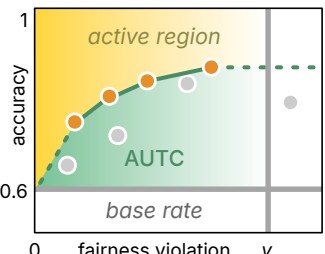

Figure 6: AUTC according to Eqs. (1) and (2), with penalty $\gamma = 1$ and violation cutoff $v$.

## C  ADDITIONAL DETAILS ON EVALUATION

For the experiments in Section 6, we used *area under the tradeoff curve* (AUTC) defined in Appendix C.1 to summarize the accuracy-fairness tradeoffs achieved by each configuration and compare their performance. The results are averaged over multiple random seeds as follows: for each seed, from the collection of classifiers obtained under varying tolerances or regularization strengths, we identify the subcollection of classifiers that lie on the Pareto frontier of the accuracy-fairness tradeoff on the validation set, that is, the classifiers for which no other achieves both higher validation accuracy and lower violation. We compute the AUTC score on this subcollection using the test set, and average the resulting AUTC scores across seeds.

In Figs. 7 to 11, we plot the tradeoff curves achieved by the classifiers from the main experiments in Section 6.1. These plots are constructed as follows. For a fixed random seed, we first generate the accuracy-fairness tradeoff curve: from the collection of classifiers trained under different tolerance settings, we remove those that are not Pareto-optimal on the validation set (those that are dominated by another with both higher accuracy and lower violation); the remaining classifiers are then sorted by their violation, and a continuous tradeoff curve is obtained by interpolating between adjacent models, spanning validation violations from $V_{\min}^{(\text{seed})}$ to $V_{\max}^{(\text{seed})}$). To aggregate tradeoff curves across random seeds and quantify uncertainty, we evaluate each curve on the *test set* at six equally spaced validation-violation percentage levels, $t \in \{0.0, 0.2, 0.4, 0.6, 0.8, 1.0\}$. At each level $t$, we identify (possibly via interpolation) the classifier whose validation-set violation is $t \cdot (V_{\max}^{(\text{seed})} - V_{\min}^{(\text{seed})})$, and record its test-set accuracy and violation. These results are then averaged across seeds at each percentage level to obtain the mean and standard deviation, which are used to plot the aggregated tradeoff curves.

### C.1  AREA UNDER THE TRADEOFF CURVE

Given a collection of (accuracy, fairness violation) pairs on the test set, obtained from a configuration under varying fairness tolerance settings of the fair algorithm (in our evaluation protocol, this collection is after filtering out pairs corresponding to tolerance settings whose validation performance is not Pareto-optimal), we compute the *area under the tradeoff curve* (AUTC) as follows; see Fig. 6 for a picture.

Let $b = \max_k \mathbb{P}(Y = k)$ denote the base rate of the problem. We compute the tradeoff curve in the following steps. (1) Filter the collection to retain only the Pareto-optimal tradeoff pairs. (2) Augment the collection with two additional points: $(b, 0)$, representing the majority-class constant classifier (which is trivially fair), and (max accuracy, $\infty$), which will extend the curve to the right. (3) Sort the resulting pairs by their fairness violations, and construct the tradeoff curve by linearly interpolating between adjacent points; let $T : [0, \infty) \to [0, 1]$ denote the resulting monotone function, which maps a violation level to its corresponding (interpolated) optimal accuracy. Finally, (4) given a penalty parameter $\gamma \in [0, \infty)$ and a cutoff violation $v \in [0, \infty)$, we compute the area under the tradeoff curve as

$$\text{AUTC} = \left( \int_0^v (v - u)^\gamma \max(0, T(u) - b) \, \mathrm{d}u \right) / \overline{\text{AUTC}}, \tag{1}$$

Table 2: Fairness violation cutoff values $v$ used in AUTC calculation (Eqs. (1) and (2)).

| Dataset | SP | TPR/(weighted-)FPR | EO |
|---------|-------|------|-------|
| Adult | 0.177 | 0.053 | 0.083 |
| ACSIncome | 0.305 | 0.374 | 0.315 |
| COMPAS | 0.220 | 0.138 | 0.218 |
| BiasBios | 0.090 | - | 0.490 |
| CivilComments | - | 0.007 | - |

where the normalization constant $\overline{\text{AUTC}}$ is the (weighted) area of the *active region* above the base rate:

$$\overline{\text{AUTC}} = \int_0^v (v - u)^\gamma \max(0, 1 - b) \, \mathrm{d}u. \tag{2}$$

The purpose of the penalty parameter $\gamma$ and cutoff violation $v$ is to prevent high AUTC scores from being dominated by models with high accuracy but poor accuracy-fairness tradeoffs in the low-violation regime (for example, see MinDiff + embeds on CivilComments dataset; Fig. 11). A larger value of $\gamma$ places more emphasis on the low-violation region, and a smaller cutoff $v$ truncates the contribution from high-violation points, which always favor high-accuracy models due to the extrapolated (max accuracy, $\infty$) entry introduced in step (2).

In our experiments, we set $\gamma = 1$ and choose $v$ as the fairness violation of the highest-accuracy classifier among all configurations using either the preds or embeds features (see Table 2).

## C.2 FAIRNESS UNDER OVERLAPPING GROUPS

On the CivilComments dataset, group membership is overlapping: each example may belong to multiple groups, so the group attribute $A \in \{0, 1\}^G$ is multi-label, where more than one (or none) of the coordinates can be 1. For any nonempty subset of groups, $I \subseteq \{1, \ldots, G\}$, $I \neq \mathbf{0}$, we define the event $\{A \in I\} = \bigwedge_{i \in I} \{A_i = 1\}$ to indicate that the example belongs to all groups in $I$ (note that it does not enforce $A_i = 0$ for $i \notin I$). For example, $\{A \in \{\texttt{christian}, \texttt{jewish}\}\}$ means that the comment mentions both Christianity and Judaism (but not necessarily only those two), and the exclusion of the empty set means that fairness is not enforced on comments that mention no religion.

To measure the disparity in FPR across all overlapping subgroups (all-way), we extend the FPR violation definition in Section 3 as:

$$V_{\text{FPR}} = \max_{\substack{I, J \subseteq [G] \\ I, J \neq \mathbf{0}}} \left| \mathbb{P}(\widehat{Y} = 1 \mid A \in I, Y = 0) - \mathbb{P}(\widehat{Y} = 1 \mid A \in J, Y = 0) \right|,$$

When working with finite samples, however, some intersections (e.g., comments mentioning many religions simultaneously) may be rare, leading to high variance in the empirical estimates. To reduce sensitivity to small subgroups, we define and use a weighted variant to report our CivilComments results:

$$V_{\text{FPR}}^{\text{weighted}} = \sum_{I \subseteq [G], I \neq \mathbf{0}} p(I, 0) \left| \text{FPR}(I) - \overline{\text{FPR}} \right|, \qquad \overline{\text{FPR}} = \sum_I \frac{p(I, 0)}{\sum_J p(J, 0)} \text{FPR}(I), \tag{3}$$

where $p(I, 0) = \mathbb{P}(A \in I, Y = 0)$, $\text{FPR}(I) = \mathbb{P}(\widehat{Y} = 1 \mid A \in I, Y = 0)$, $\overline{\text{FPR}}$ is the weighted average FPR over all nonempty subgroups.

## D FEATURIZING LLM PREDICTIONS

The final step of our framework in Section 4 involves transforming LLM predictions into a feature vector before applying the fair algorithm. We describe the feature engineering process used in our experiments below (applicable to SP, TPR, FPR, and EO fairness).

We first construct a vector of size $G \times K$ from the LLM logits, denoted by $q_{A,Y}$, that semantically encodes the LLM's estimate of the joint distribution $\log \mathbb{P}(A, Y \mid X)$. This is done as follows:

- If the prompts are designed to elicit the joint $(A, Y)$ directly, then the output logits are already in the desired format and shape, $q_{A,Y} \in \mathbb{R}^{G \times K}$.

- If the prompts assume conditional independence and predict $A$ and $Y$ separately, let $q_A \in \mathbb{R}^G$ and $q_Y \in \mathbb{R}^K$ denote the respective logits. We then construct $q_{A,Y} \in \mathbb{R}^{G \times K}$ by

$$(q_{A,Y})_{a,k} = (q_A)_a + (q_Y)_k.$$

  This ensures that

$$\mathrm{softmax}(q_{A,Y})_{a,k} = \mathrm{softmax}(q_A)_a \cdot \mathrm{softmax}(q_Y)_k.$$

- If the prompts are decomposed into predicting $Y$ and $A \mid Y = k$ for each $k$, let $q_Y \in \mathbb{R}^K$ denote the logits for $Y$, and $q_{A|Y=k} \in \mathbb{R}^G$ denote the logits for $A$ conditioned on $Y = k$. Let $\mathrm{LSE}(z) = \log \sum_i \exp(z_i)$ be the log-sum-exp function. Then we construct

$$(q_{A,Y})_{a,k} = (q_{A|Y=k})_a - \mathrm{LSE}(q_{A|Y=k}) + (q_Y)_k,$$

  so that

$$\mathrm{softmax}(q_{A,Y})_{a,k} = \mathrm{softmax}(q_{A|Y=k})_a \cdot \mathrm{softmax}(q_Y)_k.$$

In all cases, the softmax is applied after flattening.

Finally, we fit $q_{A,Y}$ to the ground-truth joint labels $(A, Y)$ using logistic regression. The predicted probabilities from this model, which lie in the probability simplex $\Delta^{G \times K}$, are then used as features for the fair algorithm.

**Overlapping Groups.** When the groups are overlapping, i.e., $A \in \{0, 1\}^G$ is multi-label, we construct the feature to represent the joint distribution over the class label $Y$ and all possible subgroup memberships. That is, $q_{A,Y} \in \mathbb{R}^{2^G \times K}$ is composed such that $(q_{A,Y})_{I,k}$ represents the prediction for $\mathbb{P}(\bigwedge_{i \in I} \{A_i = 1\}, Y = k)$, the probability that the instance has class label $k$ and belongs to all and only the groups in $I$.

For feature construction, if the prompts assume conditional independence and predict $Y$ and each $A_i$ separately (e.g., Listing 5), let $q_Y \in \mathbb{R}^K$ and $q_{A_i} \in \mathbb{R}^2$ denote the respective logits for $Y$ and $A_i$. Then we construct $q_{A,Y}$ by

$$(q_{A,Y})_{I,k} = (q_Y)_k + \sum_{i \in I} (q_{A_i})_1 + \sum_{i \notin I} (q_{A_i})_0,$$

so that

$$\mathrm{softmax}(q_{A,Y})_{I,k} = \mathrm{softmax}(q_Y)_k \cdot \prod_{i \in I} \mathrm{softmax}(q_{A_i})_1 \cdot \prod_{i \notin I} \mathrm{softmax}(q_{A_i})_0.$$

We then fit $q_{A,Y}$ to the ground-truth class labels and subgroup labels jointly—here, it is a $(2^G \times K)$-way classification task.

# E  FAIR ALGORITHMS

## E.1  REDUCTIONS

The Reductions fair classification algorithm, proposed by Agarwal et al. (2018), is based on a two-player game formulation of the fair classification problem. The algorithm relies on a cost-sensitive classification oracle and uses no-regret learning, and outputs a randomized ensemble of classifiers.

**Hyperparameters.** We use the implementation provided in the AIF360 library with default hyperparameters (Bellamy et al., 2018), and sweep the tolerance parameter for the "allowed fairness constraint violation" (eps) from $\{100, 50, 20, 10, 5, 2, 1, 0.5, 0.2, 0.1, 0.05, 0.02, 0.01, 0.005, 0.002, 0.001\}$. The base classifier is logistic regression.

### E.2 MINDIFF

MinDiff trains a real-valued classifier with a regularization term for matching its output distributions across groups (Prost et al., 2019). The distance between distributions is measured using a probability metric such as maximum mean discrepancy (MMD; Gretton et al., 2012).

We use our own implementation of MinDiff. Let $f \colon \mathcal{X} \to \Delta^K$ be a trainable mapping from inputs to class probability vectors $p_Y \in \Delta^K$, MinDiff optimizes $f$ via minimizing the joint objective:

$$\mathcal{L} = \mathbb{E}[\ell_{\mathrm{CE}}(Y, p_Y)] + \lambda R(p_Y),$$

where $\ell_{\mathrm{CE}}$ is the cross-entropy loss, $\lambda \geq 0$ is a regularization strength, and $R$ is a term that measures (and hence minimizes) the distances between the distributions of $p_Y$ conditioned across groups, depending on the fairness criterion (Section 3):

- For SP, we match the distributions of $p_Y$ conditioned on across groups $A$. If $G = 2$, then

$$R = D_{\mathrm{MMD}}(p_Y | A = 0, \ p_Y | A = 1),$$

  and for more than two groups ($G > 2$), we match each to the overall distribution,

$$R = \frac{1}{G} \sum_{a \in [G]} D_{\mathrm{MMD}}(p_Y | A = a, \ p_Y).$$

- For TPR parity (and FPR parity analogously), we only need the conditional distribution of $p_Y | Y = 1$ to be matched across groups, so for $G = 2$,

$$R = D_{\mathrm{MMD}}(p_Y | (A = 0, Y = 1), p_Y | (A = 1, Y = 1)).$$

  The case for of $G > 2$ is derived similar to above.

- For EO, building on TPR parity, in addition to matching $p_Y$ conditioned on class $Y = 1$, we also need to match that conditioned on all other classes. For $G = 2$,

$$R = \frac{1}{K} \sum_{k \in [K]} D_{\mathrm{MMD}}(p_Y | (A = 0, Y = k), p_Y | (A = 1, Y = k)),$$

  and for $G > 2$,

$$R = \frac{1}{GK} \sum_{a \in [G], k \in [K]} D_{\mathrm{MMD}}(p_Y | (A = a, Y = k), p_Y | Y = k).$$

**Overlapping Groups.** The MinDiff formulation for overlapping groups largely mirrors the case of disjoint groups, with the main difference being how the "average distribution" is defined for matching each subgroup's distribution to. Using the notation in Appendix C.2, let $\mathcal{I} \subseteq 2^{\{1, \ldots, G\}}$ denote the collection of overlapping subgroups over which fairness is enforced. We define the average conditional distribution as a weighted mixture:

$$(p_Y^{\mathrm{avg}} | Y = y) = \sum_{I \in \mathcal{I}} \frac{p(I, y)}{\sum_{J \in \mathcal{I}} p(J, y)} (p_Y | (A \in I, Y = y)),$$

where $p(I, y) = \mathbb{P}(A \in I, Y = y)$. For SP, we may define $p(I, \star) = \mathbb{P}(A \in I)$ to drop the conditioning on $Y$.

For example, under FPR parity, the regularization term becomes:

$$R = \frac{1}{|\mathcal{I}|} \sum_{I \in \mathcal{I}} D_{\mathrm{MMD}}\big(p_Y | (A \in I, Y = 0), p_Y^{\mathrm{avg}} | Y = 0\big).$$

**Hyperparameters.** We parameterize $f$ as a single-hidden-layer ReLU net of width 512 with an final softmax activation output layer, optimized using Adam with learning rate = 0.001, weight decay = 0.0, $\beta_1 = 0.9$, and $\beta_2 = 0.999$, over fives epochs with batch size 512. We sweep the regularization strength $\lambda$ over $\{10, 8, 6, 4, 3.5, 3, 2.5, 2, 1.5, 1, 0.7, 0.5, 0.3, 0.1, 0.05, 0.01, 0.0\}$ (when $\lambda = 0.0$, the training objective reduces to only minimizing the cross entropy loss without fairness constraint). Following Prost et al. (2019), we use the Gaussian kernel $k(x, x') = \exp(-\|x - x'\|^2 / \sigma^2)$ with bandwidth $\sigma = 0.1$ to compute $D_{\mathrm{MMD}}$.

### E.3 LINEARPOST

For SP, TPR, FPR, and EO fairness, LinearPost learns a linear classifier over the feature vector $p_{A,Y} \in \Delta^{G \times K}$, which represents the (estimated) joint distribution $\mathbb{P}(A, Y \mid X)$. This approach is based on a result from Xian & Zhao (2024), which shows that, under a continuity condition on the underlying data distribution (satisfied via small random perturbations to $p_{A,Y}$), provided that $p_{A,Y}$ is Bayes-optimal, the optimal fair classifier can be expressed as a linear classifier over $p_{A,Y}$. The parameters of this classifier are computed by solving a linear program, for which we use the Gurobi optimizer.

On the multi-class BiasBios dataset, because the fairness violation is dominated by disparities in group-conditional TPRs across classes, so to reduce computational overhead, we configure LinearPost to enforce multi-class TPR parity rather than full equalized odds. That is, we aim for $\mathbb{P}(\widehat{Y} = k \mid A = a, Y = k) = \mathbb{P}(\widehat{Y} = k \mid A = a', Y = k)$ for all $a, a' \in [G]$ and $k \in [K]$, while ignoring off-diagonal terms of the confusion matrix.

**Hyperparameters.** We sweep the fairness tolerance parameter $\alpha$ over 16 evenly spaced values between $\alpha_{\min} = 0.001$ and $\alpha_{\max}$, where $\alpha_{\max}$ is set to the fairness violation on the training set without any mitigation (i.e., the violation of the classifier returned by LinearPost under $\alpha = \infty$).

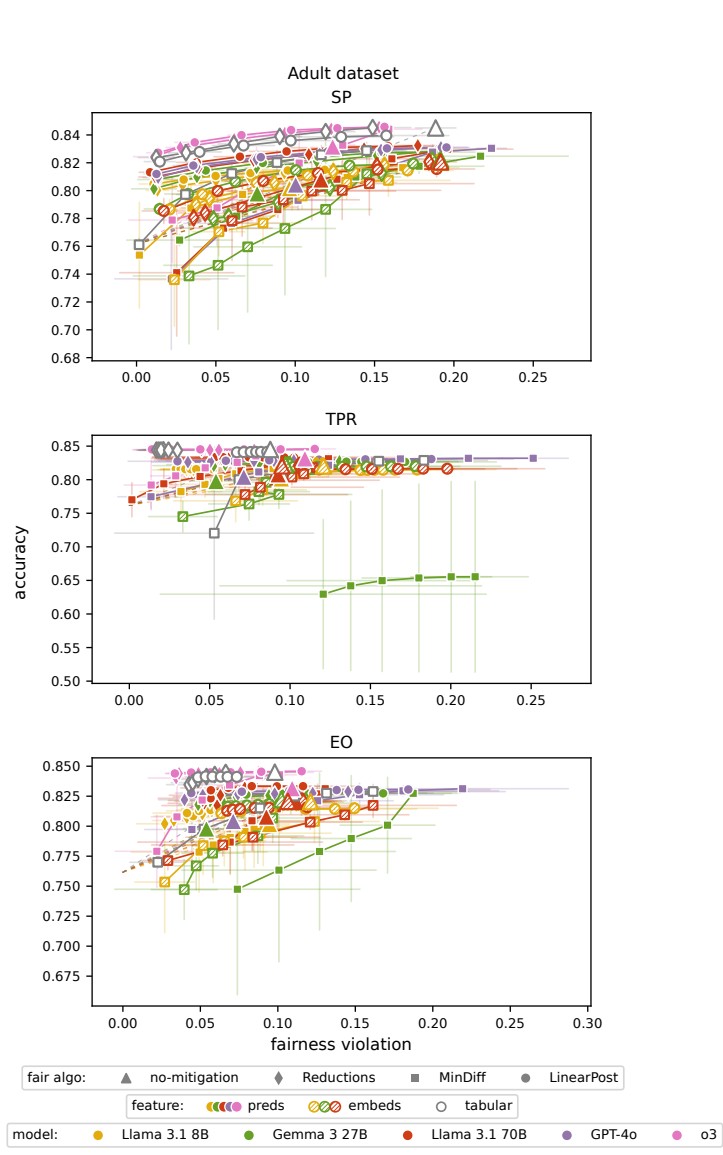

Figure 7: Accuracy-fairness tradeoffs on the Adult dataset for the experiments in Section 6.1. Dashed lines interpolate between each no-mitigation baseline and a (trivially) fair classifier that always predicts the majority class.

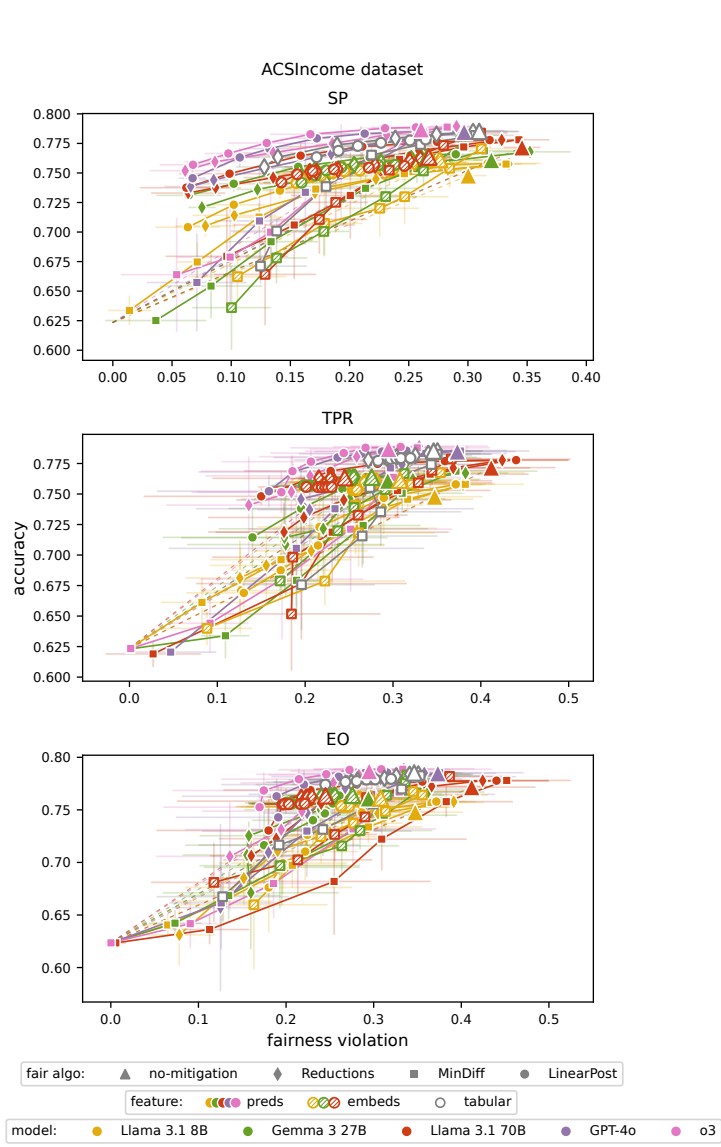

Figure 8: Accuracy-fairness tradeoffs on the ACSIncome dataset for the experiments in Section 6.1. Dashed lines interpolate between each no-mitigation baseline and a (trivially) fair classifier that always predicts the majority class.

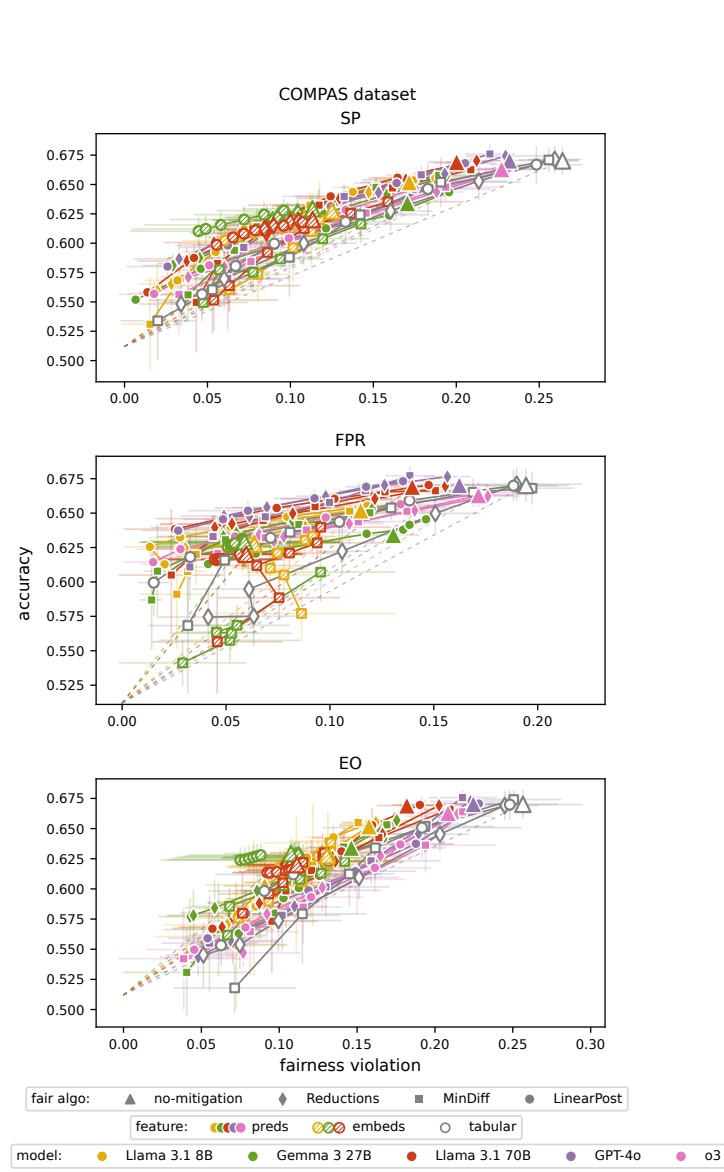

Figure 9: Accuracy-fairness tradeoffs on the COMPAS dataset for the experiments in Section 6.1. Dashed lines interpolate between each no-mitigation baseline and a (trivially) fair classifier that always predicts the majority class.

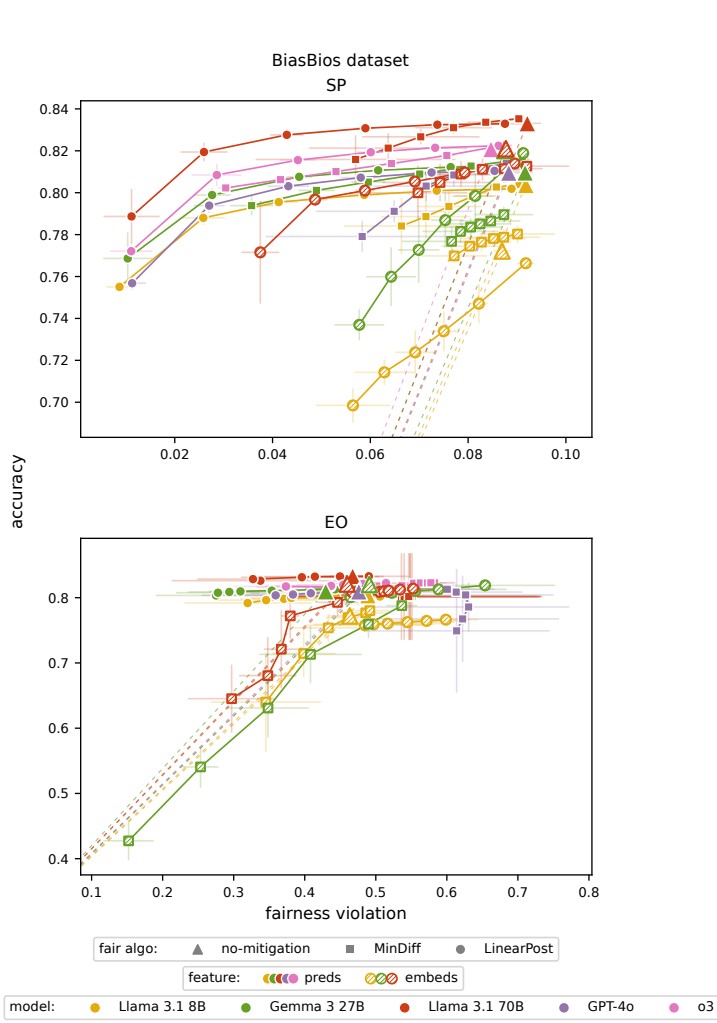

Figure 10: Accuracy-fairness tradeoffs on the BiasBios dataset for the experiments in Section 6.1. Dashed lines interpolate between each no-mitigation baseline and a (trivially) fair classifier that always predicts the majority class.

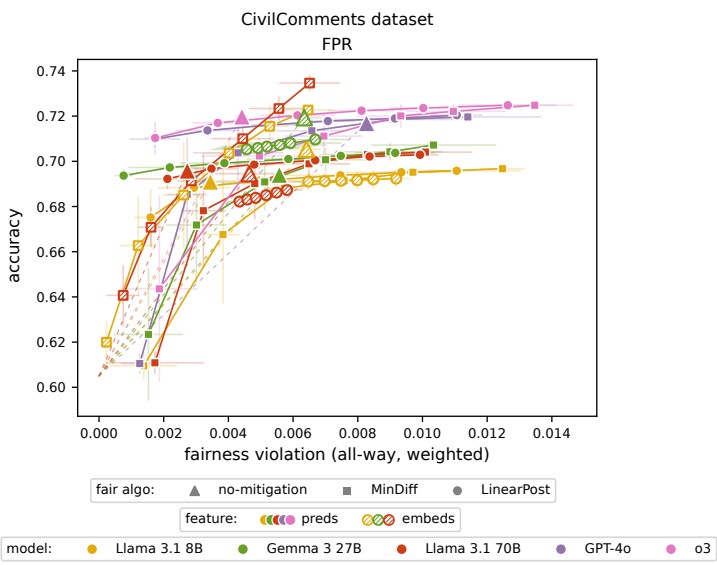

Figure 11: Accuracy-fairness tradeoffs on the CivilComments dataset for the experiments in Section 6.1. Fairness violation is computed as described in Eq. (3). Dashed lines interpolate between each no-mitigation baseline and a (trivially) fair classifier that always predicts the majority class.

Table 3: Examples from each dataset used in our experiments. For tabular datasets, the examples are shown after list serialization.

| Dataset | Example |
|---|---|
| Adult (raw encoding) | age: 39
workclass: State-gov
education: Bachelors
education-num: 13
marital-status: Never-married
occupation: Adm-clerical
relationship: Not-in-family
capital-gain: 2174
capital-loss: 0
hours-per-week: 40
native-country: United-States |
| Adult | Age: 39
Class of worker: State government employee
Educational attainment: Bachelor's degree
Education level (numeric): 13
Marital status: Never married or under 15 years old
Occupation: Administrative support and clerical workers
Relationship: Other nonrelative
Capital gain in the previous year: 2174
Capital loss in the previous year: 0
Hours worked per week: 40
Country of origin: United States |
| ACSIncome | Age: 38
Class of worker: Employee of a private not-for-profit, tax-exempt, or charitable organization
Educational attainment: Bachelor's degree
Marital status: Never married or under 15 years old
Occupation recode for 2018 and later based on 2018 OCC codes: MGR-Social And Community Service Managers
Place of birth (Recode): South Carolina/SC
Relationship: Reference person
Usual hours worked per week past 12 months: 50 |
| COMPAS | Age at the time of survey: 42
Charge degree: Felony
Charge description: Tampering With Physical Evidence
Number of prior convictions: 1
Length of stay in jail: 1 |
| BiasBios | She earned her B.A. from the State University of New York at Geneseo and M.A. and certification from the University at Buffalo. She is currently serving as GALA's Chair. Her research interests include identity development and gender within the English Language Arts classroom. She hopes to continue to develop GALA into a community of educators dedicated to how gender impacts learning. |
| CivilComments | The Philippian hymn can be interpreted various ways. It is thought to have preceded Paul and may contain the concept of the First and Second Adam theology that Paul would reference in 1 Cor and Romans. "Emptying" may refer to Jesus emptying himself of the desire to be godlike that caused Adam's downfall (Eve appears to have dropped out of the picture). |

Listing 1: Prompt templates used on the Adult dataset. For the group label prompt, the place-holder {class_condition} is replaced with either "less than or equal to 50K" or "greater than 50K" to condition the query on a (hypothetical) class label.

Template for class label (income)

```
Answer with a single letter.

Question: What is the annual income (USD) of the following
individual?
{example}

A. Less than or equal to 50K
B. Greater than 50K

Answer:
```

Template for conditional group label (sex)

```
Answer with a single letter.

Question: What is the sex of the following individual? Their
annual income (USD) is {class_condition}.
{example}

A. Female
B. Male

Answer:
```

Template for embeddings

```
Question: What is the annual income (USD) and sex of the following
 individual?
{example}

Answer:
```

Listing 2: Prompt templates used on the ACSIncome dataset. For the group label prompt, the placeholder `{class_condition}` is replaced with either "`less than or equal to 50K`" or "`greater than 50K`" to condition the query on a (hypothetical) class label.

Template for class label (income)

```
Answer with a single letter.

Question: What is the annual income (USD) of the following
individual?
{example}

A. Less than or equal to 50K
B. Greater than 50K

Answer:
```

Template for conditional group label (race)

```
Answer with a single letter.

Question: What is the race of the following individual? Their
annual income (USD) is {class_condition}.
{example}

A. White
B. Black or African American
C. American Indian or Alaska Native
D. Asian, Native Hawaiian or Pacific Islander
E. None of the above

Answer:
```

Template for embeddings

```
Question: What is the annual income (USD) and race of the
following individual?
{example}

Answer:
```

Listing 3: Prompt templates used on the COMPAS dataset.

Template for class label (recidivism)

```
Answer with a single letter.

Question: Will the following individual recidivate within two
years?
{example}

A. Unlikely to recidivate
B. Likely to recidivate

Answer:
```

Template for conditional group label (race)

```
Answer with a single letter.

Question: What is the race of the following individual? They are
{class_condition} within two years.
{example}

A. African American
B. Caucasian

Answer:
```

Template for embeddings

```
Question: Will the following individual recidivate within two
years, and what is their race?
{example}

Answer:
```

Listing 4: Prompt templates used on the BiasBios dataset. For the group label prompt, the placeholder {class_condition} is replaced with either "unlikely to recidivate" or "likely to recidivate" to condition the query on a (hypothetical) class label.

Template for class label (occupation)

```
Answer with one of the following: accountant, architect, attorney,
 chiropractor, comedian, composer, dentist, dietitian, dj,
filmmaker, interior designer, journalist, model, nurse, painter,
paralegal, pastor, personal trainer, photographer, physician, poet
, professor, psychologist, rapper, software engineer, surgeon,
teacher, yoga teacher.

Question: The following is a biography of an individual, what is
their occupation?
{example}

Answer:
```

Template for group label (sex)

```
Answer with a single letter.

Question: The following is a biography of an individual, what is
their sex?
{example}

A. Female
B. Male

Answer:
```

Template for embeddings

```
Question: The following is a biography of an individual, what is
their occupation and sex?
{example}

Answer:
```

Listing 5: Prompt templates used on the CivilComments dataset. For the group label prompt, the placeholders {religion} and {religionists} are replaced with one of the following pairs: (Christianity, Christians), (Judaism, Jewish people), (Islam, Muslims), (Hinduism, Hindus), (Buddhism, Buddhists).

Template for class label (toxicity)

```
Answer with a single letter.

Question: Is the following comment toxic (harmful, malicious,
derogatory, threat, insult, identity attack, etc.)?
{example}

A. Non-toxic
B. Toxic

Answer:
```

Template for group label (religion)

```
Answer with a single letter.

Question: Does the following comment mention {religion} or
{religionists} in any way?
{example}

A. {religion} is not mentioned
B. {religion} is mentioned

Answer:
```

Template for embeddings

```
Question: Is the following comment toxic (harmful, malicious,
derogatory, threat, insult, identity attack, etc.), and what
religion(s) does it mention in any way?
{example}

Answer:
```

