# OpenReview forum: "Group Fairness Meets the Black Box: Enabling Fair Algorithms on Closed LLMs via Post-Processing"
_ICLR.cc/2026/Conference — Submitted to ICLR 2026_

### Official Review · Reviewer_HBMh · 2025-10-22

**Soundness:** 3
**Presentation:** 2
**Contribution:** 3
**Rating:** 6
**Confidence:** 3

**Summary:**

This paper presents an interesting solution to facilitate fair outcomes from black-box LLMs for classification tasks where you don’t have access to the weights or activations to enable traditional techniques such as fine-tuning or training a classifier on embedding spaces.  The authors tackle this via prompting, treating the log prob outputs as features for probabilistic predictions.  Prompts are designed to compute priors over the output classes and likelihoods of the protected attribute conditioned on the output class, so that the joint probability of the class and protected attribute can be used as a feature for a fair classification algorithm.  The framework is tested against several datasets and popular LLMs both closed-weights and open-weights so that they can compare against algorithms that have access to internal embeddings.  They show that the algorithm does well against a no-mitigation baseline, outperforms algorithms based on embeddings and can beat algorithms trained directly on tabular data under low-data regimes.

**Strengths:**

The algorithm itself is clearly presented, particularly in figure 1 which does an excellent job of outlining the technique.

For the low data regime, this work presents an interesting solution to mitigating bias in LLM-based classifiers where you don’t have access to the weights or embeddings of the model.  This is a valuable contribution to the community.

The technique the authors use to extract the prior and conditional likelihoods does a good job of eliciting the inherent bias in the model which they directly correct with a classifier trained on the joint probability computed from these.

The experimental results are convincing, though ablations are limited.  I’d like to see more discussion on the effect of prompt variations as well as the addition of few-shot and chain-of-thought prompting as mentioned on line 476.

**Weaknesses:**

The abstract and introduction aren’t as clear as they could be as to what the algorithm is doing.  In lines 024-025 where you mention probabilistic predictions, you could mention that you using prompting to illicit the prior distribution over classes and conditional likelihoods to model the inherent bias in the model, which then form the features for the lightweight fair classifier.

Also, in the abstract you claim that your algorithm outperforms training from scratch on raw tabular features, but this is only true in the low-data regime.  You should change the abstract to reflect this.

The results presented in figure 3 are difficult to interpret.  There is no x-axis (it is just each experimental permutation), there is a mysterious clover leaf in some graphs (presumably to differentiate datasets - consider putting a box around each dataset instead), and there is just too much information in one figure.  I’d like to see these results spread out over a number of figures, each one connecting to a paragraph in section 6.1.

Some of the experimental setup that led to figure 3 is not clear (see question 3 below).

**Questions:**

1. How sensitive is the algorithm to the prompt phrasing?
2. I’m not sure subtracting the baseline from the AUTC is completely beneficial.  Without knowledge of the baseline, it is hard to tell whether the AUTC is good or not.  It is fine for comparing methods, but what if they are bad?  One case might appear fairer but inherently be lower performing due to a low baseline.  Consider updating the results to make the baseline clear or improving the text to clarify why it is OK to remove the baseline.
3. How much data did you train your classifiers on for Figure 3?  Was it the low-data regime you mention on line 452?  If so, is that a fair comparison?
4. In the conclusion you mention that you only focus on zero-shot prompting.  You should extend the work to show how you would use few-shot and chain of thought prompting for this work and what the results would be.

---

> ### Author Response · Authors · 2025-11-28
>
> We thank the reviewer for the thoughtful and helpful comments!
>
> ### W1, W2 and W3: description of the framework in the abstract and intro, and clarity re. the figures
>
> We thank the reviewer for the suggestions. In the revised draft, we (i) have made the abstract more precise in describing our framework and (ii) are explicit about our focus on the low-data regime. We have also updated the figure to improve its clarity.
>
> Regarding the layout of the plot, we deliberately chose to aggregate and display the results across all settings in the main Figure 3, as this allows for direct comparison across three dimensions that a practitioner may care about: the choice of fair algorithm, featurization strategy, and backbone LLM. For readers who prefer more granular views, we also include more detailed accuracy-fairness tradeoff curves in additional figures in the appendix.
>
> ### Q1: Prompting sensitivity
>
> We did not observe strong sensitivity to prompt phrasing, largely because we perform a calibration/refitting step (similar in spirit to Platt scaling) on the raw LLM probabilities. Prior work suggests that prompt changes often induce different offsets or scalings of the predicted probabilities while largely preserving the ranking of examples [1]; our calibration step is designed to absorb exactly these shifts and scales, which mitigates prompt-induced variability.
>
> To provide more concrete evidence, in preliminary experiments on the Adult dataset we compared two tabular data serializations schemes: one using **raw feature names** (e.g., two of the entries would be serialized as “workclass: State-gov, occupation: Adm-clerical”) and one using **natural-language descriptions** (e.g., “Class of worker: State government employee, Occupation: Administrative support and clerical workers”). Without calibration/refitting, the raw-feature serialization (likely due to dataset leakage) performed substantially better on Llama 8B, whereas after calibration this gap largely disappeared:
>
> **Adult (with raw feature names)**
>
> |  | accuracy | SP violation | TPR violation | EO violation |
> |---|---|---|---|---|
> | no refitting | 0.7579 | 0.0868 | 0.0313 | 0.0313 |
> | refitted | 0.7686 | 0.0574 | 0.0433 | 0.0433 |
>
> **Adult (with natural-language descriptions)**
>
> |  | accuracy | SP violation | TPR violation | EO violation |
> |---|---|---|---|---|
> | no refitting | 0.6675 | 0.2260 | 0.1155 | 0.1403 |
> | refitted | 0.7838 | 0.0915 | 0.0537 | 0.0537 |
>
> ### Q2: AUTC score
>
> Our AUTC score is defined analogously to AUROC. Just as, for AUROC, we only care about the area above the diagonal (assuming the curve lies above it), for AUTC we only care about the area above the `y = base_rate` line, since this baseline is achieved by a trivial constant classifier. Including the baseline would simply add a constant shift to all AUTC values and would not affect the plots or the relative rankings.
>
> ### Q3: Setup of Figure 3
>
> In Figure 3, we deliberately use a relatively small training set (exact sizes are reported in Table 1 in the appendix) to emulate low-resource scenarios where practitioners would naturally rely on zero-shot or few-shot prompting; our framework is indeed most favorable in this regime. To explore the effects of dataset size, we also report experiments in Section 6.2 where we vary the number of training examples (Figure 4), showing that training directly on raw features or on LLM embeddings (when available) only becomes competitive when substantially more labeled data are available.
>
> ### Q4: Limited models evaluated
>
> We have added a new set of experiments using OpenAI’s o3 reasoning model (closed weight), with two notable changes. First, the model performs chain-of-thought reasoning. Second, because OpenAI does not expose o3’s token log probabilities, we adopt verbal elicitation and ask the model to output probabilities directly in its generation. Our framework remains effective under this new model and setup; this both adds diversity in the evaluated LLMs and demonstrates that our approach is robust across different prompting and probability-extraction strategies.
>
> ---
>
> [1] Zhao et al., Calibrate Before Use: Improving Few-shot Performance of Language Models, 2021.

---

### Official Review · Reviewer_yy5U · 2025-10-27

**Soundness:** 3
**Presentation:** 2
**Contribution:** 2
**Rating:** 4
**Confidence:** 4

**Summary:**

This paper addresses the limitation of closed-weight large language models (such as GPT-4, Gemini, Claude, etc.) that cannot perform parameter fine-tuning, and proposes a post-processing-based framework for achieving group fairness classification under the condition of only accessing the model's prediction outputs (such as the log probabilities of tokens).
The authors view LLMs as feature extractors, obtaining the LLM's probability predictions for task labels and fairness-related variables through specific prompt design, and constructing low-dimensional features based on this. Then, they combine existing fairness classification algorithms (such as Reductions, MinDiff, LinearPost, etc.) with the existing LLMs (including GPT-4o) for training lightweight fairness classifiers. The paper conducted experiments on several datasets (Adult & ACSIncome, COMPAS, BiasBios, CivilComments) and four LLMs (including GPT-4o), and the results show that the proposed method achieves better accuracy-fairness trade-off performance in low-data scenarios.

**Strengths:**

1. The problem background holds significant practical value: The study of fairness for closed-source LLMs is a current issue of considerable practical significance, especially under the realistic constraints where model weights are inaccessible.
2. The framework has universality: The method can be compatible with various fairness algorithms and different fairness definitions, and can be applied to various types of tasks (text and tabular data).
3. The framework has strong universality: The method can be compatible with various fairness algorithms (pre-, in-, post-processing) and different fairness definitions (SP, TPR, EO, FPR, etc.), and can be applied to various types of tasks (text and tabular data).
4. The AUTC indicator was proposed to evaluate the trade-off between fairness and accuracy, which to a certain extent enhanced the quantitative comparability of the results.

**Weaknesses:**

1. The innovation is limited. The method essentially combines the existing post-processing fairness algorithm with the LLM output. The innovation is limited, especially considering the existing work [1, 2].
2. Insufficient analysis depth. The paper mainly focuses on verifying the "feasibility of the post-processing framework", but fails to conduct in-depth theoretical analysis on the mechanism for improving fairness. Besides, there is a lack of statistical significance testing for the improvement extent of fairness indicators.
3. Limited scope of application. The framework is entirely dependent on classification prediction outputs and is not applicable to generative tasks (such as text generation, summarization, fair question answering, etc.). Therefore, the assertion that "it is applicable to closed-source LLMs" should be carefully qualified in terms of scope.
4. The number of models covered in the experiment is relatively small (only 4 LLMs), and most of them are open-source models, which cannot fully verify the applicability of "closed-source LLMs".

[1] Di Gennaro F, Laugel T, Grari V, et al. Post-processing fairness with minimal changes[J]. arXiv preprint arXiv:2408.15096, 2024.
[2] Nguyen D, Gupta S, Rana S, et al. Fairness improvement for black-box classifiers with Gaussian process[J]. Information Sciences, 2021, 576: 542-556.

**Questions:**

See the above-mentioned weakness.

---

> ### Author Response · Authors · 2025-11-28
>
> We thank the reviewer for the thoughtful and helpful comments!
>
> ### W1: The innovation is limited, especially considering the existing work [1, 2]
>
> We thank the reviewer for the references; based on our reading, both works propose fair post-processing algorithms, similar to the LinearPost algorithm used in our experiments.
>
> First, we note that these works do not study applications to (closed-weight) LLMs. Second, the main contribution of our paper is to enable (and empirically evaluate) the use of fair algorithms, including those in [1, 2], on closed-weight LLMs via a prompt-based interface. In this sense, their algorithmic contributions are complementary to, and largely orthogonal to, our focus. Given their relevance, we will include [1, 2] as additional references in the final version.
>
> ### W2: Insufficient analysis depth
>
> Regarding “insufficient analysis depth,” we would like to clarify that our goal is not to introduce new fair algorithms or new theoretical guarantees, but to adapt existing, theoretically well-understood methods to the closed-weight LLM setting. The fair classification algorithms we use have already been thoroughly analyzed in the algorithmic fairness literature [3, 4, 5], and come with concrete guarantees for group fairness; we therefore deliberately build on these results rather than re-deriving their analyses. Our framework is designed to satisfy their standard assumptions: in particular, we include a calibration step so that the elicited LLM predictions meet the conditions under which post-processing methods can guarantee high group fairness. This stands in contrast to fair prompting approaches [6, 7], which treat the LLM as a black box and provide no guarantees that the model will in fact follow fairness instructions.
>
> For statistical significance, all reported results are averaged over 5 runs with different random seeds, and we include confidence intervals (one standard deviation) in all plots.
>
> ### W3: Limited scope of application
>
> We thank the reviewer for the comment and are happy to adjust the title to better reflect the scope, e.g., “Enabling Fair Classification Algorithms on Closed-Weight LLMs via Post-Processing.”
>
> We also note that we are already explicit in the abstract that our focus is on using LLMs for classification and on group fairness as the fairness notion, and our claims about applicability are all limited to the classification setting. Furthermore, in Section 2 (Related Work), we separately discuss broader fairness issues for generative LLMs, which we view as an important but distinct problem.
>
> ### W4: Limited models evaluated
>
> We have added a new set of experiments using OpenAI’s o3 reasoning model (closed weight), with two notable changes. First, the model performs chain-of-thought reasoning. Second, because OpenAI does not expose o3’s token log probabilities, we adopt verbal elicitation and ask the model to output probabilities directly in its generation. Our framework remains effective under this new model and setup; this both adds diversity in the evaluated LLMs and demonstrates that our approach is robust across different prompting and probability-extraction strategies.
>
> ---
>
> [1, 2] are as in the review
> [3] Xian and Zhao, A Unified Post-Processing Framework for Group Fairness in Classification, 2024.
> [4] Agarwal et al., A Reductions Approach to Fair Classification, 2018.
> [5] Țifrea et al., FRAPPÉ: A Group Fairness Framework for Post-Processing Everything, 2024.
> [6] Liu et al., Confronting LLMs with Traditional ML: Rethinking the Fairness of Large Language Models in Tabular Classifications, 2024.
> [7] Cherepanova et al., Improving LLM Group Fairness on Tabular Data via In-Context Learning, 2024.

---

### Official Review · Reviewer_szYd · 2025-10-31

**Soundness:** 2
**Presentation:** 3
**Contribution:** 2
**Rating:** 4
**Confidence:** 3

**Summary:**

The paper tackles group fairness for classifiers built via closed-weight LLMs. The core idea is to treat the LLM as a black-box feature extractor and elicit sufficient statistics for fair post-hoc classification purely from its probabilistic predictions. The predictions of LLMs are calibrated (via logistic regression), featurized as a low-dimensional representation of the joint distribution over (A, Y), and then passed to standard fair algorithms (Reductions, MinDiff, or LinearPost) to train a lightweight fair classifier. The experiments show that the induced classifiers yield competitive or superior accuracy–fairness tradeoffs, especially in low-data regimes.

**Strengths:**

1. The paper is clearly written, accessible, and easy to follow.
2. It addresses the important and timely issue of achieving group fairness for closed-source LLMs.
3. The experimental evaluation is comprehensive and convincing, spanning five diverse datasets (including multi-class and overlapping-group cases), four LLMs (open and closed), and three fairness algorithms.

**Weaknesses:**

1. This framework extracts P(A | Y, X) from the LLM, the extracted features inherit the model's own biases towards sensitive attributes. This means that the "sufficient statistics" are not objective measurements, but rather potentially biased surrogate indicators. Therefore, downstream calibration/post-processing may only mitigate measurement discrepancies based on these surrogate indicators without truly correcting for bias.
2. The probability extraction schemes typically require K+1 API calls per sample (one for Y, K for A | Y = k), which becomes extremely expensive and slow for common multi-class classification problems involving hundreds of classes. The limitations of closed APIs (rate limiting) exacerbate this scalability problem, and large-scale training or evaluation becomes impractical without engineering modifications or the adoption of approximation methods.
3. Experimental results show that as the dataset size increases, the performance is surpassed by embedding-based training, indicating an inherent information bottleneck in low-dimensional feature extraction. This suggests that this method is best viewed as a low-resource solution rather than a universally superior alternative, limiting its practical impact when more labeled data is available.

**Questions:**

1. Can you leverage additional information from LLMs to enrich the low-dimensional probabilistic features and improve performance in high-data regimes—for example, by incorporating chain-of-thought?
2. What is the underlying cause of model performance plateauing as training scale increases?

---

> ### Author Response · Authors · 2025-11-28
>
> We thank the reviewer for the thoughtful and helpful comments!
>
> ### W1: “Extracted features inherit model biases; ‘sufficient statistics’ are surrogate indicators”
>
> We appreciate the reviewer’s comment and share the concern that LLMs may encode social biases; we already discuss this in Section 2. Our theory assumes Bayes-optimal estimates of $P(A , Y \mid X)$, under which these probabilities are sufficient for optimal fair classification. In practice, LLM predictions need not be Bayes-optimal, potentially due to inherited social biases and, more likely, because zero-/few-shot predictions are imperfect.
>
> However, this suboptimality (potentially induced by social bias) primarily harms predictive performance, that is, how close we are to the optimal fair classifier, but does not undermine our ability to enforce group fairness. This is because we perform a calibration step on the raw LLM outputs to ensure that the estimated $\widehat P(A, Y \mid X)$ reflects the true underlying prevalences (which also mitigates the LLM’s biases), so that the post-processing algorithm receives accurate group membership information needed for achieving fairness.
>
> ### W2: “Query cost scales poorly with K+1 prompts; closed-weight models’ APIs exacerbate scalability”
>
> In our experiments, we decomposed the joint prediction of (A, Y) into separate API calls to keep each classification task simple (with fewer options), which improves performance for weaker models (in particular, the smaller open-weight LLaMA and Gemma models).
>
> However, when the LLM can handle larger label spaces, the practitioner can instead prompt directly for the joint (A, Y) label in a single call. For example, in our new experiments with OpenAI’s o3 model on CivilComments, we merged the five overlapping group labels into one joint-prediction prompt. Thus, the number of API calls per example can be reduced to one by querying (A, Y) jointly, as long as the practitioner is confident in the LLM’s ability to solve this larger classification task.
>
> ### W3 and Q2: performance plateaus with dataset size, indicating an inherent information bottleneck
>
> We agree that the observed performance plateau reflects an information bottleneck in the low-dimensional features extracted from the LLM’s predictions, and that, when one can train directly on raw features or embeddings, our method is most advantageous in low-resource settings; this is also the main takeaway from our experimental results.
>
> We would like to emphasize, however, that most commercial LLMs are closed weight, so training on embeddings is not an option. Moreover, there is growing interest in using LLMs in the zero shot and few shot regimes to build classifiers cheaply (due to data scarcity or to save labeling effort), and our framework is designed precisely for this setting. In this context, the performance plateau is naturally explained by the inherent information bottleneck in the probabilistic features, which limits the gains from increasing the amount of labeled data.
>
> ### Q1: “Can additional information improve performance?”
>
> In our updated revision, we added a new set of experiments using OpenAI’s o3 reasoning model (closed weight), which performs chain-of-thought reasoning. We observe stronger performance on the Adult, ACSIncome, and CivilComments datasets, confirming that our framework inherits performance improvements from a stronger backbone LLM. However, performance does not improve on COMPAS and BiasBios because we encounter more frequent model refusals on these datasets (in which case we default to a uniform probability, hence degrading performance): the model refuses to make predictions on sensitive queries for race and gender identities from detention profiles or biographies. While this is an attempt by the vendor to ensure fairness by refusing to give potentially unfair predictions, it also raises the interesting question of whether exceptions should be allowed when downstream users explicitly enforce fairness via post-processing.

---

### Meta-Review · Area_Chair_Mp3E · 2025-12-21

**Summary:**

The reviewers agree that the paper addresses an important and timely problem, is generally clear, and demonstrates solid empirical performance across multiple datasets, fairness definitions, and models. The approach is especially compelling in low-data regimes, where it achieves competitive or superior fairness–accuracy tradeoffs compared to embedding-based or traditional baselines.

However, concerns remain about conceptual depth, scalability, and generality. A key issue is that the extracted statistics inherit the LLM’s own biases, meaning the framework corrects bias relative to biased surrogates rather than ground truth, leaving unclear how much true fairness is achieved. The method is expensive in practice because it requires multiple API calls per instance, which becomes prohibitive for large label spaces and under API limits. Reviewers also view the novelty as somewhat incremental, largely repackaging existing fairness post-processing ideas for LLM outputs without deep theoretical analysis of why and when it works. The framework appears primarily useful only in low-resource classification settings; as data increases, embedding-based training overtakes it, suggesting limited long-term impact. Additional concerns include limited evaluation breadth for genuinely closed models, vague claims of applicability beyond classification, insufficient analysis of statistical significance, limited ablations, unclear presentation in places, and figures that are difficult to interpret.

**Reviewer Concerns:**

Concerns regarding clarification and unclear presentations have been addressed. I think the following concerns are still outstanding: The inevitable biases from the extracted features despite the calibration step; The API cost; limited novelty and applications.

**Reviewer Scores:**

4,4,6

---

### Decision · Program_Chairs · 2026-01-26

Reject